# METANETWORK: A NOVEL APPROACH TO INTERPRETING NEURAL NETWORKS

## ABSTRACT

Recent work on mechanistic interpretability, which attempts to demystify the black box of artificial neural network (ANN) models through analytical approaches, has made it possible to give a qualitative interpretation of how each component of the model works, even without using the dataset the model was trained on. However, it is also desirable from the viewpoint of interpretability to understand the ability of the entire model; and considering the previous studies on task embedding, the ability of the entire model should also be represented by a vector. In this study we propose a novel approach to quantitatively interpreting an unseen ANN's ability based on relationships with other ANNs through obtaining a low-dimensional representation of ANNs by training a "metanetwork" that autoencodes ANNs. As a first-ever attempt of such an approach, we train a "metanetwork" to autoencode ANNs consisting of one fully-connected layer. We demonstrate the validity of our proposed approach by showing that a simple k-Nearest Neighbor classifier can successfully predict properties of the training datasets of unseen models from their embedded representations.

## 1 INTRODUCTION

In recent years, the rapid growth of foundation models in size (OpenAI, 2023; Touvron et al., 2023; Anil et al., 2023) has made it more difficult than ever to grasp the abilities of artificial neural network (ANN) models. Such a situation has encouraged the study of mechanistic interpretability (Olah, 2022), which seeks to clarify the internal processing of ANNs from an analytical approach (Fig. 1). Such studies have, for example, revealed what pattern matching a certain neuron in a certain layer of a visual model performs (Olah et al., 2020), located facts about the world embedded in the parameters of fully-connected layers of a large language model, and even edited them (Meng et al., 2023). This sort of divide-and-conquer approach should undoubtedly be indispensable for a detailed understanding of the ANN's ability. However, just as it is still difficult to explain human behavior even with much of what is already known about the local mechanisms of the brain, it may be difficult to understand the essential behavior of ANNs without seeing the forest for the trees. In this regard, a concise mathematical representation of the ability of ANNs would provide a better perspective.

The ability of an ANN, which we try to understand here, and the task it is capable of performing can be regarded as two sides of the same coin. In light of prior work on task embedding, such as TASK2VEC (Achille et al., 2019), it should be possible to represent the model's overall ability in a vector as well. One drawback is that such methods assume that one has access to the dataset that the model was trained on. Given that the training process of models is often not entirely public, our goal is to quantify the ability of any ANN in a unified mathematical format, even if its task, or dataset, is unknown.

To this end, we propose in this study a novel approach to quantitatively interpreting the ANN's ability by acquiring a low-dimensional representation of an ANN through training a "metanetwork", which autoencodes ANNs (Fig. 2). We demonstrate in the Experiments section that it is indeed possible to predict an unseen ANN's ability based on distance relationships with known ANNs in the model embedding space.

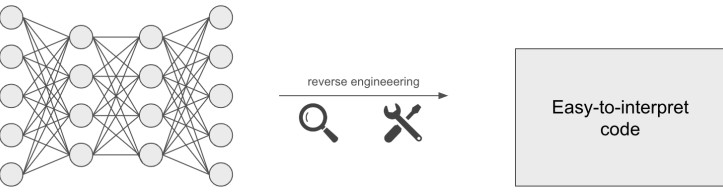

Figure 1: Mechanistic approaches to AI interpretability

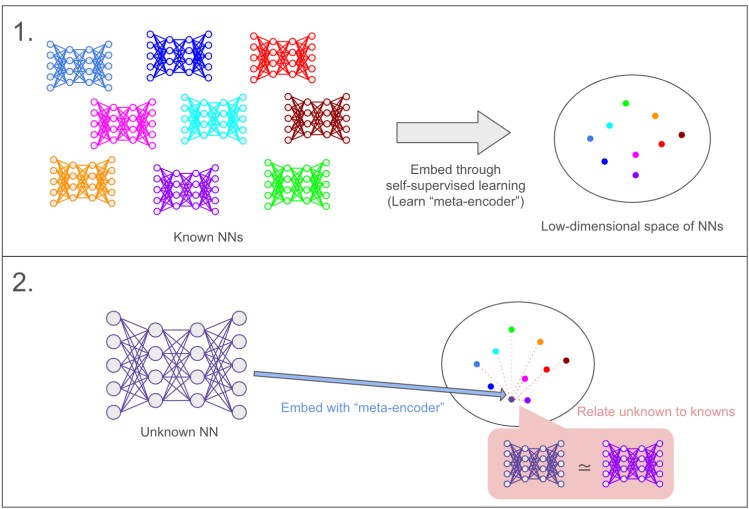

Figure 2: Our approach to AI interpretability

## 2 RELATED WORK

TASK2VEC is a method of obtaining a vector representation of a visual dataset using a common "probe network". While this study confirms that the ANN abilities can be represented in vectors, the method does not directly address the ANN representation. In the same paper, a method called MODEL2VEC is also introduced, which embeds the model in the same vector space as the task, taking into account its task performance. However, since we want to embed ANNs of any domain, not just vision, in the same space and see the relationships between them, the approach that uses modality-specific probe networks has limitations when applied to our end, not to mention that this approach needs access to the datasets.

Kornblith et al. (2019) proposes a method to calculate the similarity of representations in ANNs, and can capture small differences in internal processing of ANNs. However, since the same input data is used to compute the similarity, it would be difficult to compare models that perform different tasks.

MeLA (Meta-Learning Autoencoder) (Wu et al., 2018) is an autoencoder that obtains latent variables of an ANN model from a small number of data samples of a task and generates a model that can solve the task from those variables. Those latent variables are indeed a concise representation of the ANN model with a particular ability. Yet, the fact that the method needs some data samples to obtain the model representation makes it challenging to apply for our needs to interpret an unknown model. Nevertheless, it resonates with our motivation in that it can theoretically represent any ANN in any domain in the same format, after the data is properly preprocessed in a way that MeLA can take as input.

Futhermore, mechanistic interpretability techniques might be utilized to characterize ANNs from the abilities of its subnetworks. Some methods of feature visualization (Tyka, 2016; Mordvintsev et al., 2015; Øygard, 2015) estimate inputs that activate specific parts of the model through back propagation, which do not require the dataset the model was trained on. Although such methods are

not proposed with a motivation to represent the ability of an entire model in a unified manner, this line of approaches might turn out to be useful when our approach needs to be applied to models of various architectures.

# 3 METANETWORK

## 3.1 DEFINITION

We put the prefix "meta" to the name of an ANN when the input or/and output of the ANN is an ANN. For instance, a model that takes an ANN as input and classifies it into some categories (e.g., vision model or audio model) can be called a meta-classifier. Accordingly, MeLA is also a type of metanetwork. Later in this paper, we train a meta-autoencoder, and obtain a low-dimensional representation of an ANN with a meta-encoder, which is the encoder component of the meta-autoencoder.

## 3.2 WHAT KIND OF INFORMATION DO WE EXPECT TO BE REPRESENTED IN OUR META ENCODER OUTPUT?

We expect the output of the meta-encoder to reflect the ability of the input ANN, i.e., what task the model is capable of performing. Usually, we represent the ability of an ANN model in natural language (e.g., model A can classify handwritten digits), but such a representation not only oversimplifies the ability of the model, but also makes it difficult to compare different models.

As TASK2VEC suggests, the task performed by the model can be given a mathematical representation, and we believe that tasks are continuously distributed, including those that are difficult to express in natural language. We also believe that the model's ability should likewise be represented by a vector representation.

Here, we clarify the difference in nuance between task and ability. Indeed, the ability we are concerned with is something that is calculated individually for each model, so that even if the same task is given, the model's vector representation may vary depending on the hyperparameters used during training. However, the ability of a model is ultimately just a mapping from the input space to the output space, which should be expressed in the same style as tasks. Basically, this idea is common with MODEL2VEC and MeLA. Although our approach does not try to represent the task directly, our experimental results show that meta-encoders trained without task information embed models trained with the same task proximally.

## 3.3 WHY DO WE THINK "META-AUTOENCODER" IS PROMISING FOR INTERPRETING ANN MODELS?

As the goal is to create a tool to quantify the ability of an ANN model, the need to reconstruct the ANN is not obvious; as TASK2VEC does not bother to reconstruct the task from the task representation.

In fact, however, we ourselves do not yet know how the ability should be represented. It is true that the structure of the task space has, even though limited to the visual task, been clarified to some extent in TASK2VEC. However, the possibility that TASK2VEC approach yields to the optimal representation of the task is extremely low, since it is only a highly heuristic method.

Given this context, we believe that a much better representation can be obtained by eliminating a priori knowledge of task, or ability, and leaving the search for the optimal representation to a data-driven approach. The best choice at the moment would be to use autoencoder (Hinton & Salakhutdinov, 2006), a representation learning method that has been successful in various domains.

On the other hand, some might argue that the difficulty in interpreting metanetwork, a tool used for interpretation, is problematic. However, since a metanetwork is also an ANN, it is within the scope of the target of our approach. Since the approach is new, in this paper, we are only able to interpret a model with a simple architecture, but we believe that interpretation of metanetwork should be possible in the future as this type of research progresses.

# 4 EXPERIMENTS

We test the hypothesis that metanetwork learns a good representation of ANN model by using the model space of a meta-autoencoder trained on ANN models with the same simple architecture, and examining if it is possible to predict the ability of an unseen model with the same architecture even with a straightforward classification method.

We prepare multiple datasets for two modalities, vision and audio, and train a large number of models on each dataset. Using this model dataset, we train a meta-autoencoder. Then, we train a concatenated model of the meta-encoder with its parameters frozen and a k-Nearest Neighbor (k-NN) (Cover & Hart, 1967) classifier using the same model datasets the meta-autoencoder was trained on, combined with each model's task information, namely the dataset modality and the dataset ID. We lastly evaluate the prediction accuracy of the concatenated meta-classifier for unknown models.

## 4.1 ARCHITECTURE AND TRAINING OF MODEL TO INTERPRET

The architecture of the model to interpret is the encoder part of an autoencoder, which consists of two fully connected layers. This autoencoder has 256 dimensions for the input layer and the output layer, and the intermediate layer has 16 dimensions. For simplicity, we exclude the batch normalization and ReLU layer from the encoder part, thus the model to interpret is a 256x16 weight matrix (the encoder part has no bias). The model is trained 600 times per dataset, 500 of which are used to train the meta-autoencoder and the meta-classifier, and 100 to validate the interpretation using the meta-classifier. Weights are initialized with a glorot uniform initializer (Glorot & Bengio, 2010), the loss function is mean squared error, the number of epochs is set to 1, and Adam (Kingma & Ba, 2014) is used as the optimization function.

## 4.2 DATASET FOR MODEL TO INTERPRET

Multiple datasets are used for the two modalities, vision and audio. In order to train models of the same architecture on datasets for both modalities, we conduct the data preprocessing for each modality as follows.

### 4.2.1 VISION

All 20 classes from Pascal VOC dataset (Everingham et al.) are treated as separate vision datasets. 10 of those datasets are used to prepare the model dataset for the meta-autoencoder. For each dataset, 01-normalization and grayscaling are performed, and 1,000,000 patches in size of 16x16 are randomly sampled, each then flattened to 256 dimensions.

We refer to each dataset as "v:" followed by the first three letters of the dataset name (e.g., "v:aer" for "aeroplane" dataset in Pascal VOC dataset) in this paper.

### 4.2.2 AUDIO

All 10 classes from Urbansound8k dataset (Salamon et al., 2014), and the all 11 classes from IRMAS dataset (Bosch et al., 2014) are treated as separate audio datasets. 5 datasets from both Urbansound8k, IRMAS dataset are used to prepare the model dataset for the meta-autoencoder. For each dataset, wave data is converted to mel-spectrogram with the number of frequency bins being 16, and after 01-normalization, 1,000,000 patches in size of 16x16 are randomly sampled by moving a window of width 16 on the time axis, each then flattened to 256 dimensions.

We refer to each dataset as "a:" followed by the first three letters of the dataset name (e.g., "a:air" for "air-conditioner" dataset from IRMAS dataset) in this paper.

## 4.3 ARCHITECTURE AND TRAINING OF META-AUTOENCODER

The meta-autoencoder consists of a "unit-wise" meta-autoencoder that autoencodes each column of weights, or filter, individually, and an "inter-unit" meta-autoencoder that autoencodes the resulting set of embedded filter representations. The unitwise meta-autoencoder reshapes a 256-dimensional

filter to 16x16 first, compresses it to 3 dimensions through three convolutional layers and two full-connected layers, and then reconstructs it to 256 dimensions. The interunit autoencoder first flattens 3x16 to 48 dimensions, compresses it to 16 dimensions, and reconstructs it to 48 dimensions. These two are trained in sequence due to the need for sorting filters, which we explain below. Fig. 3 shows the training processes of the meta-autoencoder.

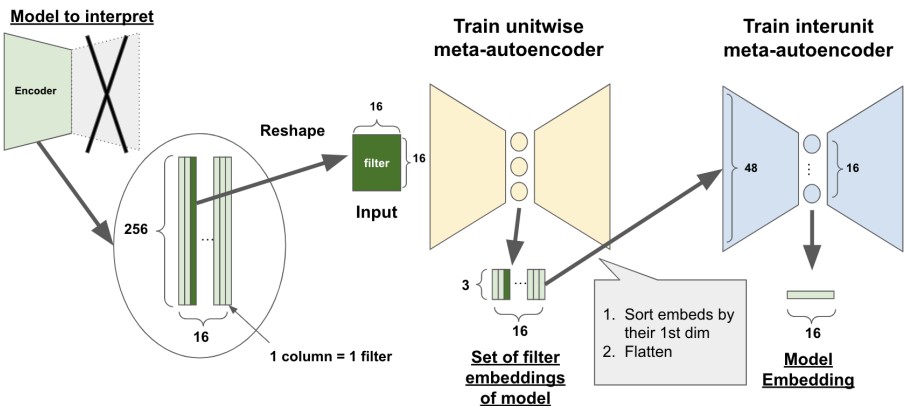

Figure 3: Training of meta-autoencoder

### 4.3.1 UNITWISE AND INTERUNIT META-AUTOENCODER

The unitwise meta-autoencoder first reshapes a 256-dimensional filter to a 16x16 first matrix. This matrix then undergoes a compression process through a series of three convolutional layers, each followed by batch normalization and max pooling. The data is then passed through two fully connected layers, and compressed to a 3-dimensional embedding space. The decoder part mirrors the encoder part. ReLU is used as the activation function throughout the network except for the output layer where the linear activation function is used.

The interunit meta-autoencoder consists of two fully connected layers with a latent space of 16 dimensions. It first flattens 3x16 to 48 dimensions, compresses it to 16 dimensions, and reconstructs it to 48 dimensions. The linear activation function is used throughout the network, with the use of bias.

For both meta-autoencoders, the weights are initialized with a glorot uniform initializer (Glorot & Bengio, 2010), the loss function is mean squared error, the number of epochs is 50, and Adam (Kingma & Ba, 2014) is used as the optimization function.

### 4.3.2 FILTER SORTING

Since the order of the filters in a fully-connected layer does not affect the ability of the entire model at all, it is appropriate to regard the 16 filters as a set. In this study, we simply consider the set of filters as an array by sorting them on the first axis (Though there are recent studies on "set" autoencoder (Janakarajan et al., 2022), we choose to not delve into it as this is not the main interest of this study). This filter sorting is conducted after the unitwise meta-autoencoder training as a preprocessing step for training the interunit meta-autoencoder.

### 4.4 INTERPRETATION OF MODELS USING K-NN CLASSIFIER

A k-NN classifier (k=10) is connected to the output layer of the meta-encoder with its parameters frozen, and the combined meta-model, which now is a meta-classifier, is fit on the training model dataset for the meta-autoencoder, with each model's training dataset ID as the label. The meta-classifier is evaluated on two levels. The first evaluation is for predicting the modality of the model's training dataset. This evaluation is conducted using 100 models for each of all 41 datasets, including those not involved in training the meta-autoencoder. The second is for predicting the training dataset ID of the model. This only uses the 20 datasets used to prepare the training model dataset for the

meta-autoencoder. To test whether obtaining a low-dimensional representation of the model actually facilitates the interpretation, a k-NN classifier that takes a model's flattened weight matrix (4096 dimensions) as input and is evaluated likewise.

## 4.5 RESULTS

### 4.5.1 PREDICTION OF DATASET MODALITY

The accuracy of modality prediction using the meta-encoder was 100% (4100/4100). This means that even the modality of the unknown task that an unseen model was trained on can be predicted with extremely high accuracy. Applying t-SNE (Van der Maaten & Hinton, 2008) (perplexity=30) to the model vector representations of the validation model datasets shows how obvious it was for the k-NN classifier to predict the modality (Fig. 4). On the other hand, the prediction accuracy without using the meta-encoder was 50% (2050/4100). The breakdown was that the modality was predicted to be visual for all models. This implies that the weight space of the models does not have a simple structure in terms of modality, which is consistent with our perception that the ANN is quite difficult to interpret as it is.

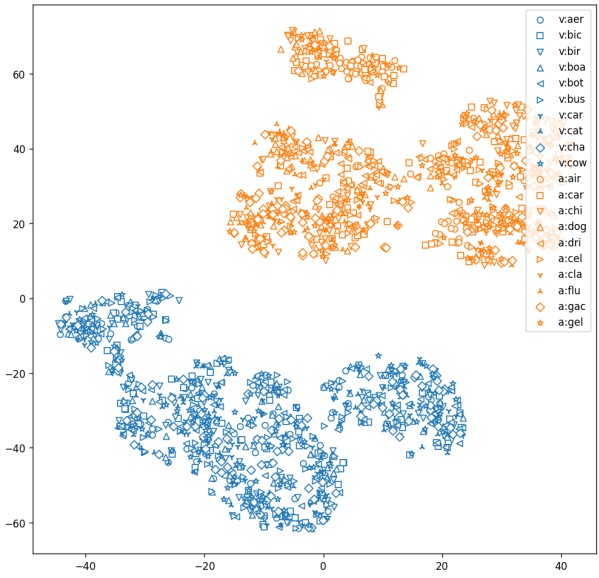

Figure 4: t-SNE visualizaiton of the model embeddings

### 4.5.2 PREDICTION OF DATASET-ID

The accuracy of dataset-id prediction using the meta-encoder was 25.4%, while the accuracy of the prediction without the meta-encoder was 7.85%. Even if the meta-classifier perfectly classifies unseen models in terms of modality, the accuracy is significantly higher than the chance level, which is 10%. Given that the task of each model is to encode a very small region of the original dataset, which is expected to be very similar in each modality, it is no exaggeration to say that the meta-encoder captures small differences between the tasks of the same modality. The confusion matrix for each prediction is Fig. 5 and Fig. 6 respectively. The diagonal element is all no less than 10% in case of using the meta-encoder.

## 5 DISCUSSION AND FUTURE WORK

### 5.1 IMPLICATIONS OF THE RESULTS

The results show that even unknown models that are difficult to interpret as-is can have their abilities inferred to some extent by using the meta-encoder, indicating that our new approach can indeed

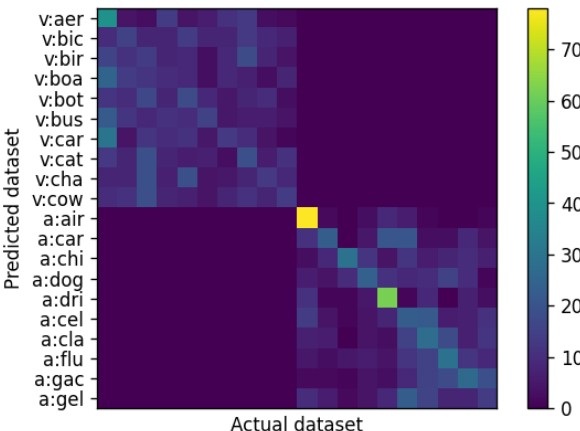

Figure 5: Confusion matrix of dataset-ID prediction with meta-encoder

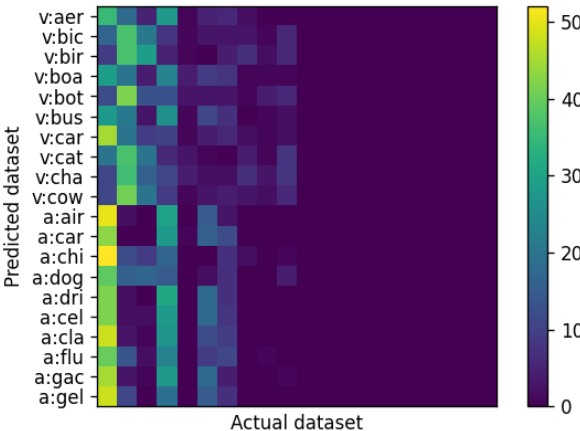

Figure 6: Confusion matrix of dataset-ID prediction without meta-encoder

contribute to AI interpretability. To obtain such results, the meta-autoencoder must have obtained a good representation of the model's ability, and the fact that we did not use any information about the model's ability when training the meta-autoencoder makes our results even more noteworthy.

## 5.2 HOW CAN WE OBTAIN A BETTER MODEL REPRESENTATION?

Although the results should be viewed positively overall, as the prediction accuracy for the dataset ID shows, there is much room for improvement.

The first area of improvement, although not the main interest of this study, is the lack of overall optimization of the meta-autoencoder. In this study, we trained a unitwise meta-autoencoder, which embeds filters, and an interunit meta-autoencoder, which embeds a set of embedded representations of filters, in sequence. However, in such a fashion, the loss of the interunit meta-autoencoder is not reflected in any parameter update of the unitwise autoencoder. Furthermore, the filter sorting method, which is a pre-processing step for training the interunit meta-autoencoder, is not very sophisticated.

The second room for improvement is to utilize knowledge about the model's ability during the meta-autoencoder training. Generally, when embedding data in an autoencoder, various hyperparameters determine how much of certain information in the input data is retained in the compressed representation. Since model parameters do not only contain information that is relevant to the ability

(e.g., the order of filters, which was eliminated intentionally in this experiment, is not relevant), in order to minimize the loss of information about the model's ability through the embedding, we may, for example, consider not only the model reconstruction loss, but also the task performance of the reconstructed model for the optimization of the meta-autoencoder. Even in this case, our goal of inferring the model's ability, whose training process is unknown, would still be served.

### 5.3 HOW CAN WE AUGMENT METANETWORK SO THAT IT TAKES IN ANY ARCHITECTURE OF MODELS?

We believe that feature visualization methods can be used to provide a unified, architecture-independent mathematical representation of the ANN's ability. As we have discussed earlier in this paper, most of the model ability we wish to grasp can be explained by the training dataset of the model. Therefore, if we infer the training dataset of the model using the feature visualization technique, without the access to the dataset we can obtain a low-dimensional vector representation of the model, using an approach similar to MeLA.

Another promising direction is converting any architecture of model into some canonical form, like a ReLU network with one hidden layer, which is proved to be possible by the well-known "Universal Approximation Theorem" (Hornik et al., 1989), and which is practically feasible (Villani & Schoots, 2023).

### 5.4 METANETWORK AND ARTIFICIAL GENERAL INTELLIGENCE (AGI)

In this subsection, we further strengthen our argument that metanetwork research should be pursued by introducing "easy-to-interpret AGI" as an application of metanetwork. Kanai et al. (2019) proposes the decoder part of a meta-autoencoder, or a meta-decoder, could be utilized to generate a model that performs a specific task that an agent is facing, and such an agent that can perform a wide spectrum of tasks can be regarded as a type of AGI. We believe that if we allow the agent to generate a model that has already been interpreted, in other words, appropriately embedded in the model space, we can say the agent is versatile while easy to interpret. The recent discovery of the scaling law (Kaplan et al., 2020) of foundation models seems to motivate AGI enthusiasts to scale up ANN models. We are sure that the AGI enabled by metanetwork would be far safer compared to other approaches.

## 6 CONCLUSION

In this study, we propose a novel approach to inferring the ability of an unknown ANN by using a low-dimensional representation of ANNs gained through training a "metanetwork" that autoencodes ANNs. As the first attempt of such an approach, we train a "metanetwork" to autoencode ANNs consisting of one fully-connected layer. We then train a k-NN classifier that predicts the model's ability. We confirmed that this classifier predicts the modality with very high accuracy and the training dataset with significantly higher accuracy than the chance level. The results show the validity of our proposed approach.

## 7 REPRODUCIBILITY STATEMENT

We ensured the reproducibility of our research by attaching all source codes used to in this study. All processes are random seeded to guarantee consistent results across different environments.

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
