# OpenReview forum: "Metanetwork: A novel approach to interpreting ANNs"
_ICLR.cc/2024/Conference — Submitted to ICLR 2024_

### Official Review · Reviewer_3Bf5 · 2023-10-27

**Soundness:** 1 poor
**Presentation:** 1 poor
**Contribution:** 1 poor
**Rating:** 3
**Confidence:** 3

**Summary:**

This paper proposes to train an autoencoder model whose input is the weights of trained DNN models. The authors empirically show, using a simple model, that it is possible to predict the task modality using the embedding space obtained by the autoencoder.

**Strengths:**

- A meta-learning approach for understanding neural networks.

**Weaknesses:**

- I could not understand the novelty of this work compared to existing works, such as TASK2VEC or MODEL2VEC mentioned in the paper, which also try to embed tasks or models.
- The datasets and models used in the experiment are too simple to have practical implications.
- Even if modality or dataset-id could be predicted by the proposed autoencoder, I'm not sure what we can say about interpreting ANNs.

**Questions:**

- In Section 4.4, it seems to me that there is nothing to train with a k-NN classifier when the meta-encoder is frozen. What do you mean by training here?

- Figure 1 suggests that the model to be interpreted is also an autoencoder. However, such an assumption is not clearly stated in the manuscript. It is also unclear why the encoder part of this model should be analyzed.

- It is unclear why we need unitwise and interunit meta-autoencoders.

- In Section 4.1: Figure -> Figure 1?

- In Section 4.3: "1 shows" -> "Figure 1 shows"?

- Formatting in references is incomplete. For example, some papers do not have a place of publication.

- (This is not a question, just a comment.) There are several works that deal directly with DNN weights as input, such as [1], which could be used to analyze trained DNN models more efficiently.

[1] Navon et al., Equivariant architectures for learning in deep weight spaces, ICML'23.

---

> ### Author Response · Authors · 2023-11-22
>
> Dear Reviewer 3Bf5,
>
> Thank you for your valuable feedback. Below we address your questions and concerns.
>
> > I could not understand the novelty of this work compared to existing works, such as TASK2VEC or MODEL2VEC mentioned in the paper, which also try to embed tasks or models.
>
> The novelty of our work is that our approach does not need access to the dataset while TASK2VEC or MODEL2VEC uses the dataset to characterize the model.
>
> > Even if modality or dataset-id could be predicted by the proposed autoencoder, I'm not sure what we can say about interpreting ANNs.
>
> We added a conceptual figure to Introduction, which illustrates the comparison between mechanistic interpretability approaches and our approach. What we really like to propose in our work is a new approach to interpreting NNs by embedding them in a lower-dimensional space and quantitatively characterizing them in relation to other NNs. In order to make this happen, the lower-dimensional space of the models has to reflect the functional similarities of models, and we demonstrate this by successfully predicting certain properties of the task that an unseen model is trained on.
>
> > In Section 4.4, it seems to me that there is nothing to train with a k-NN classifier when the meta-encoder is frozen. What do you mean by training here?
>
> Yes, you’re right that there is no parameter to update when the meta-encoder is frozen. The choice of the word “train” is indeed confusing so we changed it to “fit” in the modified version.
>
> > Figure 1 suggests that the model to be interpreted is also an autoencoder. However, such an assumption is not clearly stated in the manuscript. It is also unclear why the encoder part of this model should be analyzed.
>
> There is an explanation of this assumption in section 4.1.
>
> > It is unclear why we need unitwise and interunit meta-autoencoders.
>
> We need these two types of meta-autoencoders to facilitate ability-related encoding.
> Firstly, in our experiment setting the order of the output units does not have any functional meaning because they are in an intermediate layer in the original autoencoder. Therefore, we can take a single fully connected layer as a set of filters. Although there are some studies on the autoencoding of sets, there is no standard method. In this research, we considered a set as an array by sorting it. For example, if the original weight matrix is sorted by some key, it is possible to autoencode the entire weight end to end. However, the weight space has a very large number of dimensions, making it difficult to find an appropriate key. Therefore, we first autoencode the elements of the weight matrix, embed them in a lower dimension for feature extraction, and then sorted them using these features to create a sorting key. Thus, we first trained a unit-wise meta-encoder (for filters), sorted the obtained representations along the primary dimension, concatenated them, and then used an inter-unit meta-encoder to obtain a low-dimensional representation of the set of filter representations.
>
> > (This is not a question, just a comment.) There are several works that deal directly with DNN weights as input, such as [1], which could be used to analyze trained DNN models more efficiently.
>
> We were not aware of this work though it seems to give us some good insights on how better to embed weight matrices. Thank you so much for sharing.

---

### Official Review · Reviewer_rUtA · 2023-10-28

**Soundness:** 2 fair
**Presentation:** 1 poor
**Contribution:** 2 fair
**Rating:** 3
**Confidence:** 4

**Summary:**

- The authors approach interpretability by training a meta network, that embeds ANN in one single fully-connected layer (motivated by task embedding). Specifically, they introduce meta-autoencoders to embed ANNs in the low-dimensional latent space.
- This vector-representation of network can be used to compare different networks, e.g., to predict the modality and dataset they were trained on. The authors show that the accuracy for ability prediction, i.e., modality and dataset prediction, is superior to the one without meta network.

**Strengths:**

- The authors approach a relevant topic, i.e., network intepretability.
- The authors suggest an interesting idea, which is to find low-dimensional representations of networks that are easily comparable.

**Weaknesses:**

- The methodology section is hard to follow and very confusing. It would be helpful to have a Figure in the beginning (introduction already), to explain the intuition behind the idea and to clearly explain the role of each component (model to interpret, meta-encoder, meta-autoencoder, Knn classifier, ...")
- Figure 1 has no proper caption, which makes it hard to read and understand the figure.
- I was surprised that models are only trained for 1 epoch. Some explanation would be helpful, as this is uncommon.
- Poor experiment section: There is no comparison with other approaches. The relation of the tasks (predicting modality, predicting dataset) to intepretability is unclear.
- Nitpick: there are a couple of typos:
  - sec 2: “Those latent variables are Indeed…”
  - Sec 4.1: “(Figure)”
  - Sec 4.2.1 “the all 20 classes”
  - Sec 4.2.2 “each dataset as v:”  should be “a:”

**Questions:**

- are the original models (to be interpreted) trained on the full datasets? The dataset splitting by class label (see sec 4.2) is only done to train the meta network, right?

**Details Of Ethics Concerns:**

-

---

> ### Author Response · Authors · 2023-11-22
>
> Dear Reviewer rUtA,
>
> We really appreciate your constructive feedback. We already modified the minor formatting issues in the paper based on your review. Below we address your questions and concerns.
>
> > The methodology section is hard to follow and very confusing. It would be helpful to have a Figure in the beginning (introduction already), to explain the intuition behind the idea and to clearly explain the role of each component (model to interpret, meta-encoder, meta-autoencoder, Knn classifier, ...")
>
> We added conceptual figures to Introduction, which illustrate the comparison between mechanistic interpretability approaches and our approach. In Fig. 2, “unseen model” is the model to interpret, “meta-encoder” is what maps “unseen model” to the embedding space, and KNN classifier predicts the ability of “unseen model” by relating it to “known models”.
>
> > I was surprised that models are only trained for 1 epoch. Some explanation would be helpful, as this is uncommon.
>
> We train the models for 1 epoch because the dataset is big for the task (1 million 16x16 patches). 1 epoch turns out to be sufficient for the model to generalize well to the dataset. We believe that this is a similar situation to LLM pertaining.
>
> > Poor experiment section: There is no comparison with other approaches.
>
> We think we don’t need to compare our approach to others in our result section because the metanetwork is tested in a new problem setting where we have to infer the ability of a model without access to the dataset.
>
> > The relation of the tasks (predicting modality, predicting dataset) to interpretability is unclear.
>
> Modality and dataset ID are both properties of the task. We think that successfully predicting these is one way of interpreting NNs, especially when the original task is not known.
>
> > are the original models (to be interpreted) trained on the full datasets? The dataset splitting by class label (see sec 4.2) is only done to train the meta network, right?
>
> When we predict the modality of the task, the models to be interpreted are trained on the full datasets. But when we predict the dataset-id of the task, the models to be interpreted are trained on the datasets that are used to train the metanetwork. Please note that we trained the metanetwork without using the task information, and we only used it in the test phase.

---

### Official Review · Reviewer_Bzy7 · 2023-11-02

**Soundness:** 3 good
**Presentation:** 3 good
**Contribution:** 2 fair
**Rating:** 3
**Confidence:** 4

**Summary:**

The authors present an approach to encode the weights of a single-layer ANN. They use autoencoders to encode these weights without using the training dataset. Furthermore, they employed KNN to validate the encoding of the network weights.

**Strengths:**

1- The paper is well-written, with few formatting issues.

2- I like the idea of encoding model weights without using the training dataset, enabling work on models where access to the training data is not possible.

3- The experiments are adequate to support the claims for "very simple single-layer ANN" (though they are limited to this type of model).

**Weaknesses:**

There are several weaknesses in the paper:

1- The paper claims that encoding the model's weights improves interpretability, but it doesn't explain how it achieves this. It doesn't clarify how the model makes decisions or whether it reveals any general biases the model has learned. The only thing I can see from here is how close two models are.

2- Although the authors claim that the proposed approach can be used with any model, it appears that this may not be the case. It's unclear which layer to encode, and if we attempt to encode multiple layers, we would need different encoders due to varying input sizes. Even with the option of ReLU in section 5.3, it may be impossible for large networks. Representing an entire large network with a single-layer network doesn't seem feasible.

3- In practice, users typically train a network for a specific task they don't train hundreds of networks. Consequently, it's unclear how the encoder can be trained with very few weights.

Minor formatting issues:

1- In the related work section, change "Indeed" to "indeed" in the sentence, "Those latent variables are indeed a concise representation."

2- In Section 4.1, tag the figure number instead of writing "Figure" so that readers can identify which figure you are referring to.

3- When tagging figures, use "Fig. 1" instead of writing "[1]" for better clarity.

**Questions:**

1- Can the authors provide more detailed clarification for this part:
"Therefore, if we infer the training dataset of the model using the feature visualization technique, without access to the dataset, we can obtain a low-dimensional vector representation of the model, using an approach similar to MeLA."

2-Why is it necessary to use two different meta encoders? Can we use only one of them, such as either the "unit-wise" meta-autoencoder or the "inter-unit" meta-autoencoder only?

---

> ### Author Response · Authors · 2023-11-22
>
> Dear Reviewer Bzy7,
>
> We really appreciate your constructive feedback. We already modified the minor formatting issues in the paper based on your review. Below we address your questions and concerns.
>
> >The paper claims that encoding the model's weights improves interpretability, but it doesn't explain how it achieves this. It doesn't clarify how the model makes decisions or whether it reveals any general biases the model has learned.
>
> We added a figure in our modified version, which illustrates how the metanetwork infers a property of the task an unseen model can solve. As for general biases models have learned, we showed that the model embeddings are separated according to their task property, but we have to admit that we are yet to pinpoint what exactly it is that makes them separate in that way, and figuring this out should be included in our future work.
>
> > Although the authors claim that the proposed approach can be used with any model, it appears that this may not be the case.
>
> You are right on the point that it is quite challenging to apply our proposed methodology (unit-wise meta-encoder and inter-unit meta-encoder and so on) to any architecture of NN models. Although we didn’t empirically show this in this paper, we think we can get to embed a far wider range of NNs by first converting them to the same format of three-layer NNs as we discussed in section 5.3. But as you mentioned in the feedback, it might be difficult to be applied to “any” architecture (that uses other activation functions or has recurrent structure for example). We might have to come up with a way to solve this issue in future work.
>
> > Can the authors provide more detailed clarification for this part: "Therefore, if we infer the training dataset of the model using the feature visualization technique, without access to the dataset, we can obtain a low-dimensional vector representation of the model, using an
> approach similar to MeLA."
>
> What we mean by this sentence is that it has been shown that some feature visualization techniques (As discussed in [Olah et al. 2017](https://distill.pub/2017/feature-visualization/)) can elicit what kind of feature is captured in a particular subnetwork of a given model, without accessing the original dataset. Also, there are some work on generating a model given a dataset ([Wu et al. 2018](https://arxiv.org/abs/1807.09912); [Ha et al. 2016](https://arxiv.org/abs/1609.09106) and so on). If we combine these two together, we should be able to reconstruct the original model without using the original dataset and gain a lower dimensional representation of the model as a by-product.
>
> > Why is it necessary to use two different meta encoders? Can we use only one of them, such as either the "unit-wise" meta-autoencoder or the "inter-unit" meta-autoencoder only?
>
> We need these two types of meta-autoencoders to facilitate ability-related encoding.
> Firstly, in our experiment setting the order of the output units does not have any functional meaning because they are in an intermediate layer in the original autoencoder. Therefore, we can take a single fully connected layer as a set of filters. Although there are some studies on the autoencoding of sets, there is no standard method. In this research, we considered a set as an array by sorting it. For example, if the original weight matrix is sorted by some key, it is possible to autoencode the entire weight end to end. However, the weight space has a very large number of dimensions, making it difficult to find an appropriate key. Therefore, we first autoencode the elements of the weight matrix, embed them in a lower dimension for feature extraction, and then sorted them using these features to create a sorting key. Thus, we first trained a unit-wise meta-encoder (for filters), sorted the obtained representations along the primary dimension, concatenated them, and then used an inter-unit meta-encoder to obtain a low-dimensional representation of the set of filter representations.

---

### Official Review · Reviewer_aLkj · 2023-11-03

**Soundness:** 1 poor
**Presentation:** 1 poor
**Contribution:** 2 fair
**Rating:** 1
**Confidence:** 4

**Summary:**

The authors propose a meta network to represent the network ability to perform a task. In particular the authors propose a method to predict an unseen task of an ANN. The main motivation of the authors seems to be the task embedding approach on which the reasoning behind this work is based. Unlike to original method, the proposed method attempts to represent model's task as vector or as a result of the constructed meta network.

**Strengths:**

The concept Idea is interesting

**Weaknesses:**

I am not sure if I understood the paper correctly. The authors want to represent the hidden ability of network by a network of networks that intends to represent the component networks representation. However, the proposed model is poorly explained, no experimental data is described nor the parameters selection.

It is very hard to assess the paper results as the only representation is given using T-SNE and two feature confusion matrices representing the ability of the meta-network to distinguish between audio and video features. As such the meta network does not directly assess the ability of the specific neural network but rather simply meta-classify the modality of the features. I think this is quite different from what the authors claim in the paper.

**Questions:**

Is feature representation the same as ability representation??

---

> ### Author Response · Authors · 2023-11-22
>
> Dear Reviewer aLKj,
>
> Thank you for your valuable feedback. Below we address your questions and concerns.
>
>
> > the proposed model is poorly explained, no experimental data is described nor the parameters selection.
>
> We enriched the description of our meta-autoencoder in the modified version (See section 4.3).
>
> > As such the meta network does not directly assess the ability of the specic neural network but rather simply meta-classify the modality of the features.
> > Is feature representation the same as ability representation??
>
> We agree that feature representations of the models are not identical to ability representations, but we expect the ability information should be contained in the feature representation. We believe that the result shows even our simplistic method can get model representations that reflect the abilities well.

---

### Author Response · Authors · 2023-11-22
**Global response**

Dear reviewers,

We would like to thank the reviewers for their useful feedback to improve our paper. We modified the following parts based on your feedback.
- In response to the feedback about the lack of explanation on how our approach achieves AI interpretability, we added conceptual figures to the introduction, which illustrates the comparison between mechanistic interpretability approaches and our approach.
- In response to the feedback about the lack of explanation on metanetwork training, we enriched the explanation.
- We modified our typos, and formatting issues.

Also, we see several comments on the limitation of our method when we apply this to a broader range of model architectures. We have to admit that this might be the case at the moment as we do discuss in the paper, but we’d like to emphasize here that the main focus of our study is to propose the idea of interpreting unseen models by relating them to others in embedded space using a model autoencoder.

---

### Meta-Review · Area_Chair_dBbZ · 2023-12-08

**Metareview:**

This paper aims to provide a qualitative interpretation of different components of neural network based models based on auto-encoding (one-layer) ANNs. The resulting latent representation (via a nearest neighbor prediction function) of the autoencoder is capable of predicting what modality the original model was being trained on and predicting the accuracy of the classifier to a reasonable degree. The reviewers found this work to be an interesting study but felt it fell short in putting down a coherent picture of what was happening in the one-layer neural network that was embedded, had several design decisions for which explanations were not provided (the ANN was trained for 1 epoch) and fell short in terms of presentation.

For my own part, I was surprised to not see any mention of HyperNetworks [https://arxiv.org/abs/1609.09106, https://arxiv.org/abs/2306.06955] by the paper given the topic under study. In the future, please do situate your work with respect to this and the long line of follow up work on the same idea.

**Justification For Why Not Higher Score:**

I felt the work broached an interesting topic but omitted a vital line of related work to merit further consideration.

**Justification For Why Not Lower Score:**

N/A

---

### Decision · Program_Chairs · 2024-01-16

Reject